# Effects of Soil Nutrients on Plant Nutrient Traits in Natural *Pinus tabuliformis* Forests

**DOI:** 10.3390/plants12040735

**Published:** 2023-02-07

**Authors:** Jie Gao, Jiangfeng Wang, Yanhong Li

**Affiliations:** 1College of Life Sciences, Xinjiang Normal University, Urumqi 830054, China; 2Key Laboratory of Earth Surface Processes of Ministry of Education, College of Urban and Environmental Sciences, Peking University, Beijing 100871, China

**Keywords:** plant–soil interaction, leaf nutrient, N-limitation, *Pinus tabuliformis*

## Abstract

In light of global warming, the interaction between plant nutrient traits and soil nutrients is still unclear. Plant nutrient traits (e.g., N and P) and their stoichiometric relationships (N/P ratio) are essential for plant growth and reproduction. However, the specific role of soil nutrients in driving variation in plant nutrient traits remains poorly understood. Fifty natural *Pinus tabuliformis* forests were used as the research object to clarify the interaction between plant nutrient traits and soil nutrients. We show that: (1) The N_mass_, P_mass_ and N/P ratios of leaves were significantly higher than those of roots. The N/P ratio of both leaves and roots was less than 14. (2) Leaf nutrient traits showed diverse relationship patterns with root nutrient traits throughout the growing period. Significant changes were found in root nutrient PC2 (the second principal component of root nutrient traits) and leaf nutrient PC1 (the first principal component of leaf traits), and non-significant changes were found in other relationships between leaf and root traits (*p* > 0.05). Root nutrient traits explained 36.4% of the variance in leaf nutrient traits. (3) With the increase in soil nutrient PC2 (related to N), leaf PC2 (related to N) showed a significant trend of first decreasing and then increasing (*p* < 0.05). Only the soil N_mass_ was significantly correlated with the leaf N_mass_ (*p* < 0.05), which demonstrated that the growth and survival of *Pinus tabuliformis* forests were mainly affected by N-limitation.

## 1. Introduction

Nutrient traits refer to traits related to nutrient characteristics (e.g., N_mass_ and P_mass_), reflecting the survival strategies of plants in response to global warming, and are widely used in ecology [1,2,3]. Nitrogen (N) and phosphorus (P) are basic components of plant genetic material and nutrients, and their stoichiometric relationships significantly influence the process of plant growth and reproduction [4,5]. As a major limiting element, N is a fundamental component of enzymes [4]. Meanwhile, phosphorus drives the generation and maintenance of proteins and is also limiting in most environments [6]. The absence of nitrogen and phosphorus in leaves will affect the formation of chlorophyll, further reduce the productivity of forest communities and regulate carbon cycling [3]. The ratio of nitrogen to phosphorus (N/P) in plants reflects environmental factors, especially the nutrient supply of soil-to-plant growth [7,8]. It can clarify which elements restrict the plant’s productivity. In other words, in a habitat where P is scarce and N is relatively abundant, plant N/P is relatively high, while in the habitat where N is scarce and P is relatively rich, plant N/P is relatively low and plant P content is significantly increased [3]. Soil nutrient limitation not only affects the nutrient structure of species but also affects the composition of community species and the direction of community succession [9].

Some studies have found that climatic factors may have a certain impact on plant nutrient traits [10]. However, as the direct living environment of plants, soil provides the necessary water and nutrients for plants to survive [11]. Therefore, soil nutrient factors may play a more critical role in shaping plant nutrient trait differences. However, the specific interaction between soil nutrients and plant nutrients remains unclear.

Complex feedback regulation mechanisms exist between plant nutrients and soil nutrients [12]. For example, litter in nutrient-rich soils will introduce higher nutrients, which release large amounts of nutrients after decay, thus maintaining a higher soil fertility level. In nutrient-poor soils, plants naturally produce less litter, and their decay progresses very slowly, which further leads to soil barrenness [4,11]. N and P are important components of soil nutrients, and it is unclear whether their performance is consistent with that of plant nutrient feedback and which nutrient trait plays a more important role in plant–soil nutrient feedback.

Plants with different life forms occupy different plant–soil nutrient feedback [13]. The nutrient profiles of leaves vary significantly between different life forms due to differences in their survival strategies [2]. This plant response feedback on soil nutrient supply reflects a nutrient trade-off in plant growth and development and reflects the survival strategies adopted by plants in coping with survival pressure [14]. Based on global data, Wright et al. [15] found that the leaf nitrogen and phosphorus contents of shrubs were significantly higher than those of trees. Plants with long-living leaves have low N and P contents [2] and, thus, tree species can adapt to a living environment with low nutritional status.

In recent years, an increasing number of studies have attempted to explore the correlations between traits of different organs from the perspective of plant functional traits. Relevant studies not only help to understand the mechanisms of interaction between plant traits [4] and the utilization and allocation of resources during plant growth [16] but also have important significance in further predicting the response of plants to environmental changes. Previous studies on plant nutrient traits (e.g., N, P) often focused on aboveground organs, and few studies were conducted on roots [17]. Exploring the difference in nutrient traits between aboveground and underground organs can help us to better understand the nutrient allocation strategies of plants as well as the plant–soil nutrient feedback [18].

Songshan Nature Reserve preserves the only natural *Pinus tabuliformis* forest in North China. *Pinus tabuliformis* plays an extremely important role in resisting wind and sand, conserving water sources and purifying air, etc. The shrub species are *Syinga reticulata* var. *Mandshurica*, *Corylus mandshurica* and *Euonymus verrucosus*. *Pinus tabuliformis* is the constructive species in Songshan Nature Reserve. We aim to explore the relative effects of soil nutrients on plant nutrient traits based on the plant nutrient trait data (e.g., N_mass_, P_mass_) of roots and leaves and soil nutrient data collected from 50 natural *Pinus tabuliformis* forests. We proposed the following hypotheses: (1) There exists a significant difference between roots and leaves in nutrient traits (e.g., N_mass_, P_mass_). (2) Soil nutrient factors are better at explaining the variation in root nutrient traits than leaf nutrient traits.

## 2. Results

The N_mass_ (Figure 1A), P_mass_ (Figure 1B) and N/P ratios (Figure 1C) of leaves were significantly higher than those of roots (Table 1). We also found that the phosphorus content of trees was significantly higher than that of shrubs. However, there was no significant difference (*p* > 0.05) in nitrogen content between trees and shrubs (Appendix A).

The first two principal components (PC1 = 56.97%; PC2 = 41.54%) can explain 98.51% of the soil nutrient variation. The N/P ratio and PC1 showed a positive correlation. However, the N_mass_, P_mass_ and PC1 showed a negative correlation (Figure 2A). The first principal component mainly represents the components related to nutrient restriction. The second principal component mainly represents the components related to nitrogen.

The first two principal components (PC1 = 53.61%; PC2 = 44.22%) can explain 97.83% of the plant nutrient variation. Both the nutrient function traits of roots and the nutrient traits of leaves were positively correlated with the second principal component (Figure 2B). The first principal component mainly represents the components related to nutrient restriction. The second principal component mainly represents the components related to nitrogen.

From the results of the generalized additive models (GAMs), it is clear that the investigated leaf traits showed diverse relationship patterns with root nutrient traits throughout the growing period (Table 2; Figure 3). Significant dynamic changes were found in the root nutrient PC2 and leaf PC1, and non-significant dynamic changes were found in other relationships between the leaf and root traits. Root nutrient traits explained 36.4% of the variance in leaf nutrient traits (Table 2; Figure 4).

The soil nutrient factors had non-significant effects on the root nutrient traits (Table 2; Figure 5). The soil nutrient factors had significant nonlinear effects on the leaf nutrient traits (*p* < 0.05). With the increase in soil nutrient PC2 (related to N), leaf PC2 (related to N) showed a significant trend of first decreasing and then increasing (*p* < 0.05). Generally speaking, soil nutrients explained 25% of the variance in the leaf nutrient traits.

## 3. Discussion

We found that the N_mass_ and P_mass_ of leaves were significantly higher than those of roots. Leaves are the main organ for photosynthesis, and N_mass_ and P_mass_ are closely related to protein synthesis [2]. Plant roots cannot participate in photosynthesis due to lack of chlorophyll. Therefore the N_mass_ and P_mass_ of plant roots are significantly lower than those of leaves [19,20,21,22,23,24,25].

We also found that the P_mass_ of tree leaves was significantly higher than that of shrub leaves. Plants of different life forms have unique niches and different resource utilization strategies for light, temperature and water under environmental pressure [2,25,26,27,28,29,30,31,32]. However, there was no significant difference in N_mass_ between tree and shrub leaves. The “community construction theory” based on nutrient traits explains that, in a local community, competition may lead to divergence between traits, but habitat screening may lead to the convergence of traits [33]. Often, the habitat selection effect causes different species to form more consistent characteristics so as to adapt to the same environment [3].

There exists a tradeoff between the nutrient traits of different plant organs [15]. N_mass_ and P_mass_, in plant leaves, are mainly used for photosynthesis, while those of roots are mainly used for underground ecological processes so as to adapt to adverse environments. Therefore, when plants absorb nutrients from the soil, they will balance the nutrients according to their environment [34]. When plants are in an environment with sufficient resources (sufficient light, water, and heat), they will use more nutrients for photosynthesis to maximize resource utilization and facilitate plant growth and reproduction [26]. When plants are affected by osmotic stress, they will use more nutrients in underground processes (e.g., rooting) to avoid the threat [9].

In the past century, aboveground ecology has attracted extensive attention. However, the ecological links between aboveground and underground components remain unclear [2,3]. This knowledge gap hampers our ability to understand and predict the comprehensive responses of an ecosystem to environmental stresses [2,3]. Increasing evidence emphasizes that the importance of strong interactions between aboveground and underground components in regulating ecosystem multifunctionality and responses to global change [19,20,21,22,23,24,25,26].

Plants and soil participate in the global material cycle together, existing in a close relationship [27]. Plants absorb nitrogen and phosphorus from the soil through their roots and return them to the soil in the form of litter [20]. Therefore, there is a feedback relationship between soil and plant nutrients [3]. The aboveground element characteristics of plants are usually related to the soil nutrient content. As the main substrate for plant growth, soil contains organic matter, nitrate nitrogen and ammonium nitrogen, which are decomposed to continuously provide essential nutrients for the normal physiological activities of plants. This enables the soil and plant to achieve and maintain a balanced element ratio through the dynamic exchange of nutrient supply and demand [28,29,30,31,32]. We found that only soil N_mass_ was significantly correlated with leaf N_mass_, offering evidence that the growth and development of *Pinus tabuliformis* forests were mainly limited by the supply of soil N_mass_, and P_mass_ was not the key element limiting factor.

## 4. Materials and Methods

### 4.1. Study Area

The longitude and latitude range of Songshan Nature Reserve is 115°43′44″ E–115°50′22″ E, 40°29′9″ N–40°33′35″ N. The annual average temperature is 8.5 °C, the highest temperature in the hottest month is 39 °C, and the lowest temperature in the coldest month is −27.3 °C. The annual average duration of sunshine is 2836.3 h, the annual average frost-free period is approximately 150 days, the annual average rainfall is 493 mm, and the annual average evaporation is 1770 mm. The reserve has the second highest peak in Beijing, with a maximum altitude of 2198.39 m. Most mountains measure between 600 and 1600 m. There are three types of soil connected with the elevation changes: brown forest soil, mountain brown soil and mountain meadow soil. Mountain meadow soil is mainly distributed under shrub vegetation above an altitude of 1800 m. The reserve is rich in animal and plant resources, including 713 species of higher wild vascular plants, more than 300 species of medicinal plants and 158 species (subspecies) of birds. Fifty representative plots (30 m × 30 m) were established in natural *Pinus tabuliformis* forests. The average elevation is 800 m. The maximum altitude is 875 m, and the minimum altitude is 770 m. The soil type is brown forest soil.

### 4.2. Nutrient Trait Data

More than 10 mature and well-developed *Pinus tabuliformis* trees were selected from each plot to collect fresh (one-ye ar-old) needles and twigs. The collected samples were mixed evenly and placed into paper file bags. We selected roots with diameters greater than 2 mm for our research. The contents of N (%) and P (%) in the leaves were determined after sterilization at 105 °C, drying at 60 °C and mechanical grinding. The average value of each sample was taken to calculate the average contents of N (%) and P (%). Soil samples of the surface layer (0–20 cm) were collected under the selected tree. The soil samples were mixed fully and evenly. After air-drying in the laboratory, impurities were removed and the contents of N (g/kg) and P (g/kg) were determined after grinding and screening with 0.25 mm mesh. Roots of *Pinus tabuliformis* collection were carried out by a root-tracking method. The main roots of the sampled *Pinus tabuliformis* were found first, and the fine roots on the main roots were sequentially exposed by gradually removing sediment downward in the direction of the main roots. Roots containing at least five grades were cut off with pruning scissors, and these fine root samples were placed in self-sealing bags for preservation through temporary freezing. After taking the root samples back to the laboratory, the soil attached to the root samples was washed with water. The samples were graded according to the root order. The measured fine roots were placed in an oven at 60 °C for 72 h to maintain their weight. Root samples were kept for testing after crushing and screening with a 2 mm sieve. The total N content was determined by Kjeldahl determination, the leaf P content was determined by the molybdenum antimony anti-colorimetric method, and the soil total phosphorus was determined by the alkali fusion-Mo-Sb anti-spectrophotometric method [19,20,21,35]. The calculation method for the N_mass_ and P_mass_ of the leaves and roots of shrub species is the same as that for *Pinus tabuliformis.*

### 4.3. Data Analysis

Principal component analysis (PCA) was used to reduce the dimensions of the plant nutrient traits, root nutrient and soil nutrient factors and was conducted within the R environment using the “vegan” package.

Generalized additive models (GAMs) were used to evaluate the effects of soil nutrient factors on leaf and root nutrient traits. This approach utilizes both parametric and non-parametric components to reduce the model risks inherent to linear models [22]. The model can be summarized as:(1)$$g(E(Yi))\;=\;\beta_{0}\;+\;S_{1}(x_{i})\;+\;S_{2}(x_{i})\;+\;e_{i}\;$$
where g is a link function, *E*(*Y*_i_) is the estimate for the responsible variable *Y*_i_, S_1_ is the smooth function of x_i_ for different light treatments, and S_2_ is the smooth function of x_i_ throughout the investigation time. x_i_ (i = 1, 2, 3,…, 12) are the explanatory variables, and they are the number of new rhizomes, new rhizome length, new rhizome diameter, etc. β_0_ is the constant term and e_i_ is the error term. All calculations were conducted within the R environment using the “mgcv” package.

## Figures and Tables

**Figure 1 plants-12-00735-f001:**
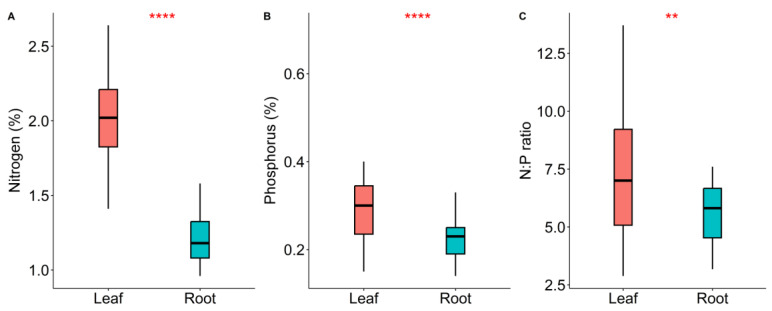
A comparison of the differences in the N_mass_ (**A**), P_mass_ (**B**), and N/P (**C**) of different organs (leaf and root). * represents *p* < 0.05, ** represents *p* < 0.01, **** represents *p* < 0.0001.

**Figure 2 plants-12-00735-f002:**
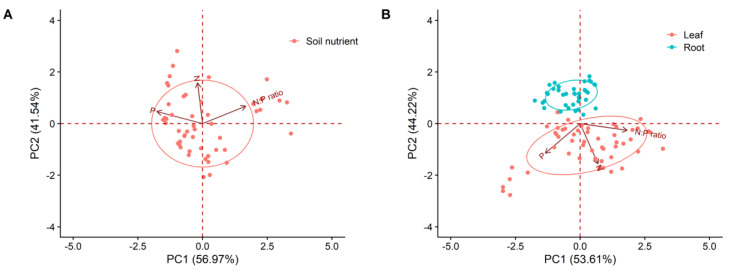
Principal component analysis (PCA) of nutrient characteristics of (**A**) soil and (**B**) leaves and roots. Nutrient characteristics included the N_mass_, P_mass_ and N/P ratio.

**Figure 3 plants-12-00735-f003:**
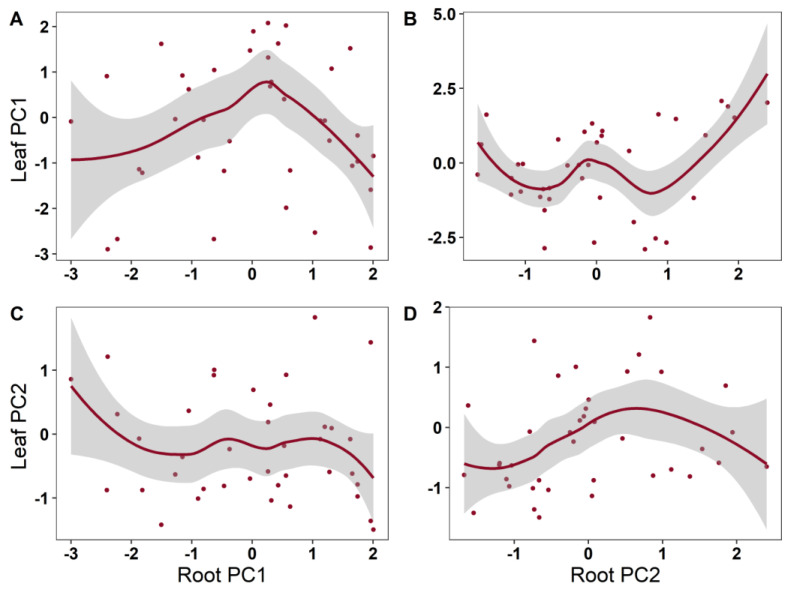
The plots of the GAMs smooth function indicating the effects of root nutrient traits on leaf nutrient traits. (**A**): root PC1 vs. leaf PC1; (**B**): root PC2 vs. leaf PC1; (**C**): root PC1 vs. leaf PC2; (**D**): root PC2 vs. leaf PC2.

**Figure 4 plants-12-00735-f004:**
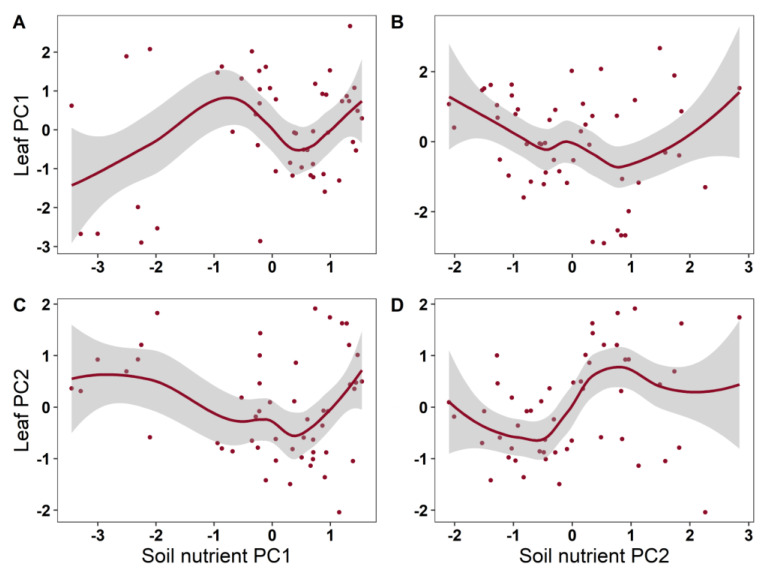
The plots of the GAMs smooth function indicating the effects of soil nutrient factors on leaf nutrient traits. (**A**): soil nutrient PC1 vs. leaf PC1; (**B**): soil nutrient PC2 vs. leaf PC1; (**C**): soil nutrient PC1 vs. leaf PC2; (**D**): soil nutrient PC2 vs. leaf PC2.

**Figure 5 plants-12-00735-f005:**
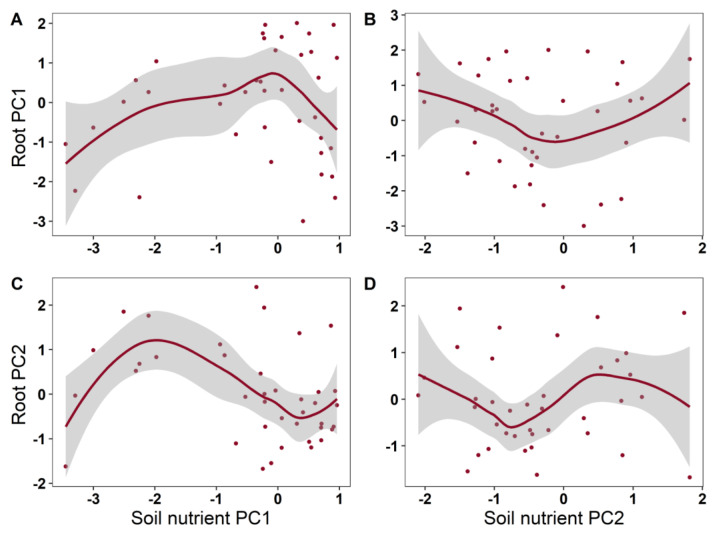
The plots of the GAMs smooth function indicating the effects of soil nutrient factors on root nutrient traits. (**A**): soil nutrient PC1 vs. root PC1; (**B**): soil nutrient PC2 vs. root PC1; (**C**): soil nutrient PC1 vs. root PC2; (**D**): soil nutrient PC2 vs. root PC2.

**Table 1 plants-12-00735-t001:** Variations in the N_mass_, P_mass_ and N/P ratio of leaves, roots and soil.

	Nutrient Characteristics	Mean Value (g/g)	Coefficient of Variation (%)	Max–Min (g/g)
Leaf	N_mass_	2.03	3.81	1.23
	P_mass_	0.32	6.02	0.56
	N:P	7.38	10.59	5.82
Root	N_mass_	1.19	2.15	0.62
	P_mass_	0.22	1.23	0.20
	N:P	5.6	29.56	4.42
Soil	N_mass_	3.22	41.72	5.38
	P_mass_	0.43	7.74	0.62
	N:P	8.46	20.05	6.4

**Table 2 plants-12-00735-t002:** Results of the generalized additive models (GAMs) explaining the influence of soil nutrient and root nutrient traits on leaf and root nutrient traits. * *p* < 0.05.

Parameters	Independent Variable	Degrees of Freedom	F Value	Pr (>|t|)	R^2^_adj_
Leaf PC1	Soil PC1	3.328	1.496	0.227	0.163
	Soil PC2	1.970	1.452	0.217	
Leaf PC2	Soil PC1	1.432	0.498	0.6806	0.25
	Soil PC2	5.484	2.282	0.0475 *	
Root PC1	Soil PC1	1.797	1.729	0.190	0.108
	Soil PC2	1.641	0.790	0.451	
Root PC2	Soil PC1	3.086	2.757	0.0511	0.181
	Soil PC2	1.000	0.231	0.6340	
Leaf PC1	Root PC1	2.843	1.939	0.1352	0.364
	Root PC2	4.418	2.802	0.0337 *	
Leaf PC2	Root PC1	1.000	0.392	0.536	0.0861
	Root PC2	1.927	1.740	0.153	

## Data Availability

Not applicable.

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
