# Peer review of "Effects of Soil Nutrients on Plant Nutrient Traits in Natural Pinus tabuliformis Forests"

_plants, 2023, doi:10.3390/plants12040735_

Round 1

Reviewer 1 Report

Soil nutrients play an important role in regulating plant aboveground and underground processes. The authors explored the effects of soil nutrients on the aboveground and underground functional properties of natural Chinese pine forests, which is of great botanical value. This research has abundant data, reasonable research methods and rich research results. As this research mainly focuses on plant functions, it is very consistent with the theme of this special issue. Therefore, I suggest publishing it in the journal Plants after minor revision.

1:line 91 should be Nmass, Pmass.

2:line 123 : Please remove the equivalent and replace it with no significant difference.

3:line 172 and line 176, Please revise the format of references.

4:The discussion part of the article can be appropriately strengthened, focusing on the interaction process of aboveground and underground ecology.

Author Response

Dear Editor and Reviewers,

We  really  greatly  appreciate  you  for  processing  our  manuscript  entitled “Effects of Soil Nutrients on Plant Nutrient Traits in Natural Pinus tabulaeformis Forests” (Manuscript ID: plants-2173315). We are  grateful for your and the reviewers’ valuable suggestions and comments on the manuscript. We have polished the whole manuscript based on the comments, in order to ensure that it is clear and as brief as possible, following the Plants-Basel format.

The point-by-point responses to your and the reviewers’ comments can be  found below. The comments are shown in black font and the responses are shown in blue font.

Yours sincerely,

Jie Gao and coauthors

Reviewer #1 (Remarks to the Author):

Soil nutrients play an important role in regulating plant aboveground and underground processes. The authors explored the effects of soil nutrients on the aboveground and underground functional properties of natural Chinese pine forests, which is of great botanical value. This research has abundant data, reasonable research methods and rich research results. As this research mainly focuses on plant functions, it is very consistent with the theme of this special issue. Therefore, I suggest publishing it in the journal Plants after minor revision

Response: Thank you very much for your recognition and valuable suggestions of our work.

1:line 91 should be Nmass, Pmass.

2:line 123 : Please remove the equivalent and replace it with no significant difference.

3:line 172 and line 176, Please revise the format of references.

4:The discussion part of the article can be appropriately strengthened, focusing on the interaction process of aboveground and underground ecology.

Response: Thank you very much for your valuable suggestions. We revised the line 91 and changed N,P to Nmass and Pmass.  Line 123 have been changed as: Our results show that the average Nmass of natural Pinus tabulaeformis forests have no significantly difference with the global average level (2.01%) and the average Pmass is higher than the global level (0.199%). We have revised the format of line 172 and line 176. We have also revised the discussion. In the past century, aboveground ecology has attracted extensive attention. However, the ecological links between aboveground and underground components remains not clearly[2-3]. This knowledge gap hampers our ability to understand and predict the comprehensive responses of ecosystem to environmental stresses. Increasing evidences emphasize that the importance of strong interactions between above- and below-ground in regulating ecosystem multifunctionality and the responses to global change [20-27].

Reviewer 2 Report

English is not my native language, so I don't judge the quality of the language. Although in terms of botanical morphology, needles are leaves, it would be less misleading to use the term needles. Thus, it is not clear in which cases it is written about all the trees of the stand, including bushes, where only about pines and needles and pine roots. Which species belong to the group of shrubs, the leaves of deciduous trees and the needles of conifers should be analyzed separately. It is also necessary to formulate more precisely in the headings of figures and tables which of the species of groups of trees and shrubs has been analyzed, what data is included, if you do not read the entire article it is not clear. Are the trees just pine, or are there other trees? What proportions are they grown in? As a supplementary, only one figure is included, it is more efficient to place it in the results section of the article.

Author Response

Dear Editor and Reviewers,

We  really  greatly  appreciate  you  for  processing  our  manuscript  entitled “Effects of Soil Nutrients on Plant Nutrient Traits in Natural Pinus tabulaeformis Forests” (Manuscript ID: plants-2173315). We are  grateful for your and the reviewers’ valuable suggestions and comments on the manuscript. We have polished the whole manuscript based on the comments, in order to ensure that it is clear and as brief as possible, following the Plants-Basel format.

The point-by-point responses to your and the reviewers’ comments can be  found below. The comments are shown in black font and the responses are shown in blue font.

Yours sincerely,

Jie Gao and coauthors

Reviewer #2 (Remarks to the Author):

English is not my native language, so I don't judge the quality of the language. Although in terms of botanical morphology, needles are leaves, it would be less misleading to use the term needles. Thus, it is not clear in which cases it is written about all the trees of the stand, including bushes, where only about pines and needles and pine roots. Which species belong to the group of shrubs, the leaves of deciduous trees and the needles of conifers should be analyzed separately. It is also necessary to formulate more precisely in the headings of figures and tables which of the species of groups of trees and shrubs has been analyzed, what data is included, if you do not read the entire article it is not clear. Are the trees just pine, or are there other trees? What proportions are they grown in? As a supplementary, only one figure is included, it is more efficient to place it in the results section of the article.

Response: Thank you very much for your valuable suggestions. There were only one tree species (Pinus tabulaeformis) in our study. Natural Chinese pine forests are generally pure forests, and there are basically no other tree species. The shrub species under the natural Chinese pine forest are usually rare. According to your opinion, we listed shrub species in the method section of the article. Shrub species are Syinga reticulata var. Mandshurica, Corylus mandshurica and Euonymus verrucosus.

Reviewer 3 Report

A brief summary

A review of the manuscript entitled: „Effects of Soil Nutrients on Plant Nutrient Traits in Natural Pinus tabuliformis Forests” is partly interesting and promising. I recommend significant revisions before the consideration for possible publication in the Plants.

Major concerns

My first concern is the usage of some uncommon term (nutrient traits) without any interpretation, which makes common readers of this journal hard to read. This term is mental shortcut. The beeter term is leaf nitrogen and phosphorus content sensu Wright et al 2004 as a part leaf economic spectrum and root economic spectrum.

Another major concern on the structure of Introduction, especially hypotheses. H1 - division into forms of shrubs and trees is not well described in the methods and results. H2 – trade-off, this hypothesis has not been discussed (see Discussion section). L. 78 ‘different plant organ’ – leaf and root, other organs have not been tested.

Specific comment

mistakes in names: Pinus tabulaeformis according to The Plant List (http://www.theplantlist.org/) = Pinus tabuliformis – change in whole text.

Table 1 – column Range (Max- Min) – this is not a range?. Column ‘Mean Value’ 1.91, on the box-plot A value ~ 1.2, please clarify.

Author Response

Dear Editor and Reviewers,

We  really  greatly  appreciate  you  for  processing  our  manuscript  entitled “Effects of Soil Nutrients on Plant Nutrient Traits in Natural Pinus tabulaeformis Forests” (Manuscript ID: plants-2173315). We are  grateful for your and the reviewers’ valuable suggestions and comments on the manuscript. We have polished the whole manuscript based on the comments, in order to ensure that it is clear and as brief as possible, following the Plants-Basel format.

The point-by-point responses to your and the reviewers’ comments can be  found below. The comments are shown in black font and the responses are shown in blue font.

Yours sincerely,

Jie Gao and coauthors

Reviewer #3 (Remarks to the Author):

A review of the manuscript entitled: „Effects of Soil Nutrients on Plant Nutrient Traits in Natural Pinus tabulaeformis Forests” is partly interesting and promising. I recommend significant revisions before the consideration for possible publication in the Plants.

Response: Thank you very much for your valuable suggestions. We have carefully revised the MS according to your suggestions.

My first concern is the usage of some uncommon term (nutrient traits) without any interpretation, which makes common readers of this journal hard to read. This term is mental shortcut. The beeter term is leaf nitrogen and phosphorus content sensu Wright et al 2004 as a part leaf economic spectrum and root economic spectrum.

Response: Thank you very much for your suggestions. Nutrient traits refer to traits related to nutrient characteristics, which are widely used by a large number of ecologists(e.g., .

  • J Durán, A Rodríguez, JM Fernández-Palacios, et al. Changes in leaf nutrient traits in a wildfire chronosequence[J]. Plant & Soil, 2010, 331(s1-2):69-77.
  • Liu, JX, Zhang, et al. Changes in leaf nutrient traits and photosynthesis of four tree species: effects of elevated [CO2], N fertilization and canopy positions[J]. J PLANT ECOL-UK, 2012, 2012,5(4)(-):376-390.
  • Wang R ,  Wang Q ,  Zhao N , et al. Different phylogenetic and environmental controls of first‐order root morphological and nutrient traits: Evidence ofmultidimensional root traits[J]. Functional Ecology, 2018.
  • Magudeeswari P ,  Sastry E ,  Devi T R . Principal component (PCA) and cluster analyses for plant nutrient traits in baby corn {Zea mays L.)[J]. Indian journal of agricultural research, 2019(3):53.)

In order to avoid causing some misunderstanding, we have added an explanation to it in the introduction section and thank you very much.

Another major concern on the structure of Introduction, especially hypotheses. H1 - division into forms of shrubs and trees is not well described in the methods and results. H2 – trade-off, this hypothesis has not been discussed (see Discussion section). L. 78 ‘different plant organ’ – leaf and root, other organs have not been tested.

Response: Thank you very much for your valuable suggestions. We have added the descriptions of shrubs. Shrub species are Syinga reticulata var. Mandshurica, Corylus mandshurica and Euonymus verrucosus. The calculation method of Nmass and Pmass in leaves and roots of shrub species is the same as that of Pinus tabulaeformis.

We have revised the hypotheses according to your suggestions. We proposed the following hypotheses: (1) There exists significant difference between roots and leaves in nutrient traits. (2) The ability of soil nutrient factors in shaping the variation of root nutrient traits is stronger than that of leaf nutrient traits.

mistakes in names: Pinus tabulaeformis according to The Plant List (http://www.theplantlist.org/) = Pinus tabuliformis – change in whole text.

Table 1 – column Range (Max- Min) – this is not a range?. Column ‘Mean Value’ 1.91, on the box-plot A value ~ 1.2, please clarify.

Response: Thank you very much for your valuable suggestions. We modified the name of Pinus tabulaeformis. We have also revised the mean value of 1.19, not 1.91. The column Range (Max- Min) now is Max- Min.

Round 2

Reviewer 3 Report

The Authors provide sufficient answers to the questions. The Authors have made the appropriate changes in the main text and table. My comment was understood the other way around. The previous version of the manuscript had two names for the same species Species name needs correction.

The correct species name:

The Plant List (http://www.theplantlist.org/) - Pinus tabuliformis

Author Response

We have revised the name of pinus tabuliformis. Thanks for your suggestion.